# Fluid–Structure Interaction Analysis of Perfusion Process of Vascularized Channels within Hydrogel Matrix Based on Three-Dimensional Printing

**DOI:** 10.3390/polym12091898

**Published:** 2020-08-24

**Authors:** Shuai Yang, Jianping Shi, Jiquan Yang, Chunmei Feng, Hao Tang

**Affiliations:** 1School of Electrical and Automation Engineering, Nanjing Normal University, Nanjing 210023, China; shuaiyang96@163.com (S.Y.); 61013@njnu.edu.cn (C.F.); tang.hao96@126.com (H.T.); 2Jiangsu Key Laboratory of 3D Printing Equipment and Manufacturing, Nanjing Normal University, Nanjing 210042, China

**Keywords:** three-dimensional bioprinting, vascularized channels, perfusion pressure, hydrogel concentration, fluid–structure interaction, crosslinking density

## Abstract

The rise of three-dimensional bioprinting technology provides a new way to fabricate in tissue engineering in vitro, but how to provide sufficient nutrition for the internal region of the engineered printed tissue has become the main obstacle. In vitro perfusion culture can not only provide nutrients for the growth of internal cells but also take away the metabolic wastes in time, which is an effective method to solve the problem of tissue engineering culture in vitro. Aiming at user-defined tissue engineering with internal vascularized channels obtained by three-dimensional printing experiment in the early stage, a simulation model was established and the in vitro fluid–structure interaction finite element analysis of tissue engineering perfusion process was carried out. Through fluid–structure interaction simulation, the hydrodynamic behavior and mechanical properties of vascularized channels in the perfusion process was discussed when the perfusion pressure, hydrogel concentration, and crosslinking density changed. The effects of perfusion pressure, hydrogel concentration, and crosslinking density on the flow velocity, pressure on the vascularized channels, and deformation of vascularized channels were analyzed. The simulation results provide a method to optimize the perfusion parameters of tissue engineering, avoiding the perfusion failure caused by unreasonable perfusion pressure and hydrogel concentration and promoting the development of tissue engineering culture in vitro.

## 1. Introduction

Since Wilson and Boland first proposed three-dimensional (3D) bioprinting technology in 2003, this field has received increasing attention from the scientific community [1]. 3D bioprinting can produce complex multimaterial structures with controllable shape structures and controllable material components [2]. Therefore, 3D bioprinting technology has enabled us to push the development of complex tissue constructs for in vitro applications even further by setting up microfluidic channels within the printed organ equivalents for perfusion and the possibility of vascularization [3,4,5,6]. The in vitro culture of tissue engineering requires simultaneous growth of the internal vascular system to facilitate the exchange of oxygen, nutrients, growth factors, and metabolic wastes between cells and the internal environment [7,8]. In order to create vascularized channels in a hydrogel matrix, there are three main strategies in the field of bioprinting: extrusion-based, drop-based, and laser-based bioprinting. Each has been used in a variety of biological applications, providing different performance in terms of cell viability, deposition rate, print resolution, scalability, cost, or material compatibility [9].

Impressive progress has been accomplished in fabricating complex tissue constructs in the past few years. For example, Griffith et al. fabricated a vascularized liver on a small scale by using the inkjet printing technique. They were pioneers in investigating the role of scaffold architecture from biodegradable polyesters and culture conditions for achieving hepatic function in long-term perfusion cultures [10]. In 2013, Zein et al. printed the first human liver along with its complex network of vascular and biliary structures. Specifically, successful 3D synthetic livers were printed which replicated the native livers of six patients, three living donors, and three respective recipients. These results demonstrate the potential efficacy of 3D-printed tissue engineering and organs with a vascular network in the human body as a substitute for treating partially or irreversibly damaged tissue [11]. A complex liver organoid was precisely printed using a stereolithographic bioprinting approach by Tobias Grix et al. The liver equivalents were designed with hollow channels to allow for perfusion of the organoid. The printed liver tissue equivalents were found to have higher albumin and cytochrome P_450_ 3A4expression over a two-week cultivation period, when compared to monolayer controls. Tight junction protein zonula occludens-1and multidrug resistance-associated protein 2expression remained stable in the printed tissue [12]. 3D bioprinting has been considered as a promising method to address the increased demand for tissues or organs with long-term mechanical and biological stability, suitable for transplantation.

Hydrogel is superior to other materials in construction for tissue engineering and can simulate the body environment well because of its high moisture content and high elastic modulus [13]. In our previous work, we printed sacrificial ink into the hydrogel matrix to fabricate perfusable vascularized channels within the hydrogel matrix [14]. Our approach is summarized in Figure 1. Hydrogel was first printed layer-by-layer and then exposed to ultraviolet light with a wavelength of 405 nm, which formed a self-supporting matrix with the grooves of the internal channels. When the desired height of hydrogel was achieved, the piezoelectric nozzle was switched to eject sacrificial ink within the printed hydrogel layer to fill the internal grooves. Once the printing of the internal structure was finished, hydrogel was extruded to wrap the whole structure. Then, it was immersed in a CaCl_2_ solution to fully crosslink the structure, and the sacrificial ink was then removed to form 3D hollow vascularized channels.

The difficulty in further culturing the printed tissue engineering in vitro is how to provide sufficient oxygen and nutrients to ensure the growth of the internal cells [15]. Diffusion of nutrients is the most commonly used method for in vitro culture tissue engineering. However, due to the diffusion limitation, cells at a depth of more than 100 μm cannot obtain sufficient nutrition, which is not conducive to culturing tissue engineering in vitro [16]. Perfusion culture can provide a flowing and fresh culture medium for tissue engineering, which can not only provide nutrients for cells but also take away the metabolites. It is an effective means of enhancing cell vitality and promote cell growth [17]. Therefore, perfusion of culture medium into the vascularized channels within tissue engineering is the most effective method for tissue engineering culture in vitro [18,19].

Thus, it is extremely important to set the perfusion pressure reasonably in the perfusion process of tissue engineering with a pressure perfusion device. If pressure is too low, the fluid flow rate will be insufficient, and the corresponding material transport rate will not be provided; if pressure is too high, the fluid flow rate will be too fast, which will cause the greater pressure on the vascularized channels and larger deformation of the vascularized channels, even the fracture of tissue engineering. In addition to the external perfusion pressure, it is also necessary to consider the mechanical properties of hydrogel with different concentrations and crosslinking density, which will also affect the perfusion effect. For the perfusion culture of tissue engineering, in this work, we proposed a fluid–structure interaction (FSI) simulation method for user-defined hydrogel tissue engineering with embedded vascularized channels to optimize the perfusion parameters. The results show the effects of perfusion pressure, hydrogel concentration, and crosslinking density on the flow velocity, pressure on the vascularized channels, and deformation of vascularized channels. The proposed FSI simulation method can optimize the parameters of tissue engineering perfusion culture so as to promote the development of tissue engineering culture in vitro.

## 2. Materials and Methods

The FSI simulation analysis includes two parts: hydrodynamic analysis of blood flow velocity and the pressure on the vessel; structural mechanics analysis of the vessel under the pressure. The pressure on the vessel calculated in the fluid simulation was loaded into the inner wall of the vessel to calculate the deformation of the vessel. Because the deformation was relatively small, the influence of the deformation of the vessel negligibly affected the fluid. Therefore, the impact of the fluid characteristics on the pressure and deformation of the blood vessel was analyzed using a one-way multiphysical field coupling method. For the convenience of accurate calculation and analysis, the viscosity temperature characteristic of blood was not considered, that is, blood adopted a constant dynamic viscosity to make the model have a better convergence. The fluid dynamics model was solved by the transient laminar flow method, while the mechanical model was solved by quasi-steady state solid mechanics.

### 2.1. Model

COMSOL Multiphysics 5.4 (COMSOL, Shanghai, China) was used to establish the FSI finite element model and carry out the simulation. The model is presented in Figure 2. Figure 2a shows the overall model. The size of the hydrogel matrix: 24 mm × 24 mm × 20 mm. The size of the internal vascularized channel: large radius of 8 mm, small radius of 1 mm, vertical pitch of 8 mm, and 2 turns. Figure 2b shows the size of the vessel wall.

### 2.2. Fluid Model

At different Reynolds number, the fluid flow state is different, and the resistance of the object is also different. At first, the Reynolds number was calculated to determine the state of blood flow in the vascularized channel. The Reynolds number in the vascularized channel can be described by the following equation.
(1)$$R_{e}=\frac{v_{b}\cdot D\cdot \rho }{\mu }$$
where $v_{b}$ is the velocity of the fluid; *ρ* is the density of the fluid; *μ* is the dynamic viscosity of the fluid and *D* is the channel diameter. $v_{b}$, *ρ*, *μ*, and *D* can be selected as 2 mm, 0.10 m/s, 1060 kg/m^3^, and 0.005 Ns/m^2^, respectively, with the reference to blood physiological parameters. According to Equation (1), the Reynolds number obtained by calculation is less than 2300, so the fluid flow characteristic can be considered as laminar flow. The governing equations for incompressible, laminar, Newtonian fluid flows are Navier–Stokes equations as below [20].
(2)∇U=0
(3)$$\rho \cdot (U\cdot \nabla )U=\nabla \left(-p+\mu \cdot \left(\nabla U\;+\;{\left(\nabla U\right)}^{T}\right)\right)$$
where ***U*** is the velocity of the fluid and ***p*** is the pressure. In the fluid domain, the inner wall of the vessel was assumed to be a nonslip boundary. The inferior cerebral vein with a diameter of 2 mm can offer the reference of blood physiological parameters for the simulation model. The average blood pressure of the cerebral vein in the supine position 11 mmHg (1500 Pa) was applied at the outlet as constant pressure [21]. Perfusion pressure at the inlet was set as 12, 13, 14, and 15 mmHg, respectively, for FSI simulation. The results can be analyzed to optimize the perfusion pressure.

### 2.3. Solid Model

When blood flows through the vessel, it exerts pressure on the inner wall, causing the vessel to deform. Because the external hydrogel matrix has a supporting effect on the vessel, it restricts the deformation of the vessel to a certain extent. Thus, the restraint of the external hydrogel matrix on the vessel needs to be considered in the mechanical model. The mechanical model of the vessel is based on solid mechanics and superelastic material model, and coupled with the results of fluid mechanics analysis to analyze solid mechanics. The physical equations of solid materials can be expressed as follows.
(4)$$\{\begin{array}{c} \sigma =(\frac{W}{\varepsilon }\cdot I+\nabla U)\\ \;\varepsilon =\frac{1}{2}\left[{\left(\nabla U\right)}^{T}\;+\;\nabla U+{\left(\nabla U\right)}^{T}\nabla U\right]\;\\ W=\frac{1}{2}\mu \left(I_{1}-3-2In\left(J_{e1}\right)\right)\;+\;\frac{1}{2}\lambda {\left[In\left(J_{e1}\right)\right]}^{2} \end{array}$$
where σ is stress, ε is strain, W is strain energy, and $J_{e1}$ is the proportion of the elastic deformation to the total deformation. For the vessel, the density, Lamé parameter *μ*, and Lamé parameter *λ* were respectively considered to be 960 kg/m^3^, 6.20 × 10^6^ N/m^2^, and 1.24 × 10^8^ N/m^2^. For the hydrogel matrix, the density, Lamé parameter *μ*, and Lamé parameter *λ* were respectively considered to be 1050 kg/m^3^, 2.758 × 10^3^ N/m^2^, and 2.4827 × 10^5^ N/m^2^ [22].

## 3. Results and Discussion

### 3.1. The Effect of Perfusion Pressure in the Perfusion Process

#### 3.1.1. The Simulation Results of Fluid Flow Velocity

In Figure 3, the simulation results of the flow velocity under different perfusion pressure are presented. The results demonstrate that the flow velocity is uniformly distributed and varies along the radial direction under different pressure. The flow velocity reaches its maximum at the center of the radial section and decreases along the radial direction to 0 near the vessel wall. In the radial direction of a circular pipe, the relationship between the axial velocity of the fluid and the distance to the center of the pipe cavity is as follows.
(5)$$v=v_{max}\left[1-{\left(d_{r}lr\right)}^{2}\right]$$
where r is the radius and $v_{max}$ is the maximum speed. The results are in accordance with Equation (5) and the characteristics of human blood velocity. The results demonstrate that the flow velocity is proportional to the perfusion pressure. When the perfusion pressure increases from 12 mmHg to 13, 14, and 15 mmHg, the fluid flow velocity at the center increases from 45 × 10^−3^ m/s to 0.09, 0.14, and 0.18 m/s, respectively. Because the reference value of blood flow velocity of the cerebral vein is 9.9 ± 1.4 cm/s, the simulation result meets the requirements when the perfusion pressure at the inlet is 12 mmHg. The results show that when the perfusion pressure is not appropriate, the flow velocity will be too slow or too fast, and the flow velocity generated will not be conducive to the tissue engineering culture in vitro.

#### 3.1.2. The Simulation Results of the Pressure on the Vessel

The simulation results of the pressure on the vessel wall under different perfusion pressure are shown in Figure 4. In the perfusion process, the boundary load of fluid mechanics exerts pressure on the inner wall of the vessel. The results reveal that the average pressure on the vessel significantly increases with increasing perfusion pressure. Moreover, the maximum pressure on the vessel wall is at the inlet, and gradually decreases along the outlet direction, thus forming a pressure gradient and generating the fluid flow force. The venous pressure in the human body is very low, ranging 0~2660 Pa (0~19.9 mmHg) [23]. When the perfusion pressure is 13 mmHg, satisfying the venous flow rate, the maximum pressure at the inlet is 14.83 mmHg, about 1960 Pa, which not only meets the physiological parameters of the human body but is also far less than the compressive strength of the hydrogel matrix.

#### 3.1.3. The Simulation Results of the Deformation of the Vessel

The deformation of the vessel under different perfusion pressure is shown in Figure 5. The pressure applied to the vessel wall can cause the vessel to deform. It can be observed in Figure 5 that the overall deformation is mainly the expanded deformation along the radial direction, and the deformation degree under each perfusion pressure is not significant. The deformation of the vessel has a similar change trend with the pressure. The maximum pressure at the entrance leads to the maximum deformation, and the minimum pressure at the exit leads to the minimum deformation.

### 3.2. The Effect of Hydrogel Concentration in the Perfusion Process

Sodium alginate hydrogel is widely used in the fabrication of tissue engineering because of its excellent biocompatibility, water solubility, and safety. The concentration of sodium alginate in the hydrogel matrix can affect its mechanical properties. Therefore, in the simulation analysis of the perfusion process, the impact of external perfusion pressure on the perfusion effect should be considered as well as the impact of the concentration of hydrogel on the perfusion effect [24]. When the inlet pressure is set as 13 mmHg and the outlet pressure as 11 mmHg, the Lamé μ and Lamé λ were selected as 1.207 × 10^4^ and 1.0862 × 10^5^ N/m^2^, 1.5511 × 10^4^ and 1.3966 × 10^5^ N/m^2^, 1.965 × 10^4^ and 1.769 × 10^5^ N/m^2^, 2.758 × 10^4^ and 2.4827 × 10^5^ N/m^2^, and the corresponding sodium alginate hydrogel concentrations were 20%, 30%, 40%, 50% [25].

The relationship between the concentration and the flow velocity, pressure, and deformation was studied by simulation. The results indicate that the hydrogel concentration has little effect on the flow velocity and vessel wall pressure. At different concentrations, the maximum flow velocity at the center of the vessel and the maximum pressure at the inlet are consistent, and are respectively 0.09 m/s and 12.88 mmHg. However, the concentration has a significant influence on the deformation. The simulation results are shown in Figure 6. It can be observed that when the perfusion pressure is the same, the deformation decreases with the increase in concentration, and the relationship between the two is inversely proportional. The reason is that the increase in the hydrogel concentration will lead to the increase in its elastic modulus, making it not easy to deform. The simulation results are in agreement with the theoretical basis [26]. Combining the printing parameters requirements of tissue engineering and human physiological parameters for the model, when the perfusion pressure is selected as 13 mmHg, the deformation with the concentration of 20–30% is more consistent with the requirements.

### 3.3. The Effect of Crosslinking Density in the Perfusion Process

In the case of hydrogel used in tissue engineering, in addition to the concentration of hydrogel, the crosslinking density is also widely used to modulate the elasticity or to improve the mechanical properties of hydrogel. At the same time, hydrogel degradation can also affect the perfusion effect. However, the degradation rate can be adjusted by controlling crosslinking density. The greater the crosslinking density of hydrogel, the longer the degradation time and the smaller the degradation degree [27,28,29,30]. Therefore, it is necessary to discuss the effect of crosslinking density on perfusion. Sodium alginate hydrogel are crosslinked by immersing in a CaCl_2_ solution. Different calcium ion concentrations will lead to different crosslinking densities, which will further affect the elastic modulus of the material. The Lamé μ and Lamé λ were selected as 1.1379 × 10^4^ and 1.0241 × 10^5^ N/m^2^, 1.3793 × 10^4^ and 1.2414 × 10^5^ N/m^2^, 1.5517 × 10^4^ and 1.3655 × 10^5^ N/m^2^, 1.7241 × 10^4^ and 1.5517 × 10^5^ N/m^2^, and the corresponding calcium ion concentrations were 0.025, 0.05, 0.075, and 0.1 mol/L [25].

The results indicate that crosslinking density has little effect on the flow velocity and vessel wall pressure, which is the same as the concentration. Similarly, crosslinking density has a significant influence on deformation. The simulation results are shown in Figure 7. The results reveal that the deformation decreases with increasing calcium concentration. For instance, the deformation is 7.8 μm for 0.025 mol/L and 6.4 μm for 0.05 mol/L. The simulation results are in accordance with the theoretical basis. The reason is that the increase in calcium concentration will lead to the increase in crosslinking density of hydrogel, which will further increase the elastic modulus and make the hydrogel matrix difficult to degrade and deform [31].

## 4. Conclusions

In the previous work, tissue engineering with the internal vascularized channel was fabricated. In order to successfully culture tissue engineering by perfusion in vitro, a hydrogel tissue engineering model was established, and the FSI finite element analysis of the perfusion process was carried out. The simulation mainly studied the effect of perfusion pressure, hydrogel concentration, and hydrogel crosslinking density on the perfusion process. Results show that when the concentration and crosslinking density of hydrogel are constant, as the perfusion pressure increases, the average fluid flow velocity increases and the pressure on the vessel increases, leading to an increase in the deformation of the vessel. When the perfusion pressure is constant, the flow velocity and pressure are basically unchanged with the increase in the concentration and crosslinking density of hydrogel, but the deformation of the vessel decreases. The reason is that the increase in the concentration and crosslinking density of hydrogel leads to an increase of the elastic modulus, which makes the material difficult to deform. In this work, we propose a simulation method, which can optimize the perfusion parameters for user-defined tissue engineering to avoid the failure of tissue engineering perfusion culture due to unreasonable perfusion pressure setting so as to promote the development of tissue engineering culture in vitro.

## Figures and Tables

**Figure 1 polymers-12-01898-f001:**
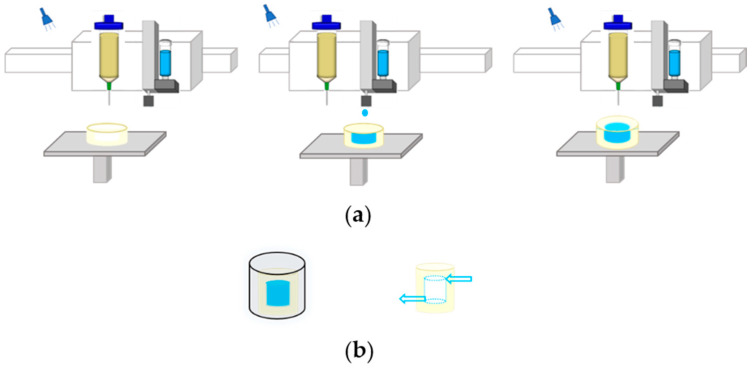
Schematic showing the process of fabricating the hydrogel matrix with embedded perfusable channels: (**a**) sequential printing of ejecting a sacrificial ink into photocurable hydrogel matrix by dual-head printer; (**b**) post-printing process including immersion in a CaCl_2_ solution to crosslink and remove sacrificial ink to fabricate channels.

**Figure 2 polymers-12-01898-f002:**
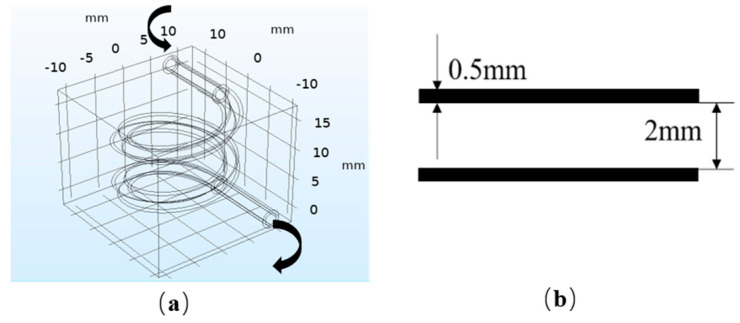
Fluid–structure interaction (FSI) model: (**a**) overall model; (**b**) vascular model.

**Figure 3 polymers-12-01898-f003:**
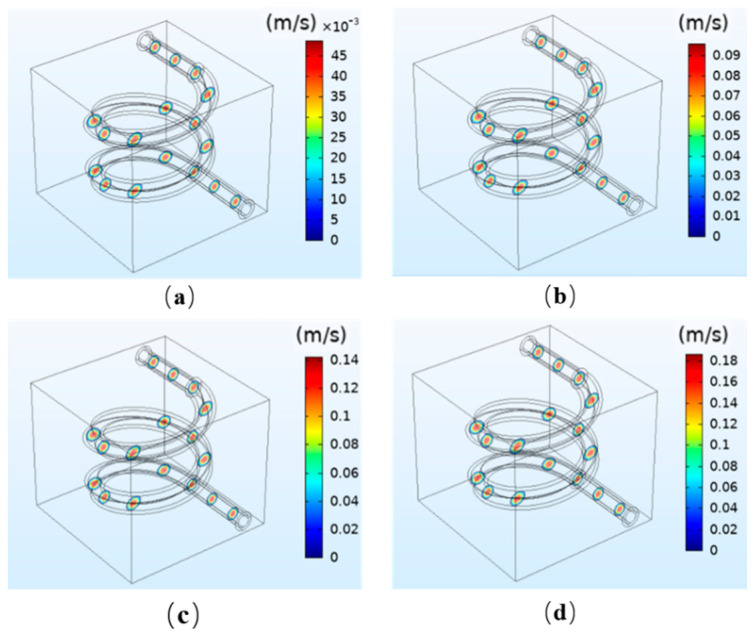
Simulation results of the fluid velocity under different perfusion pressures: (**a**) 12 mmHg; (**b**) 13 mmHg; (**c**) 14 mmHg; (**d**) 15 mmHg.

**Figure 4 polymers-12-01898-f004:**
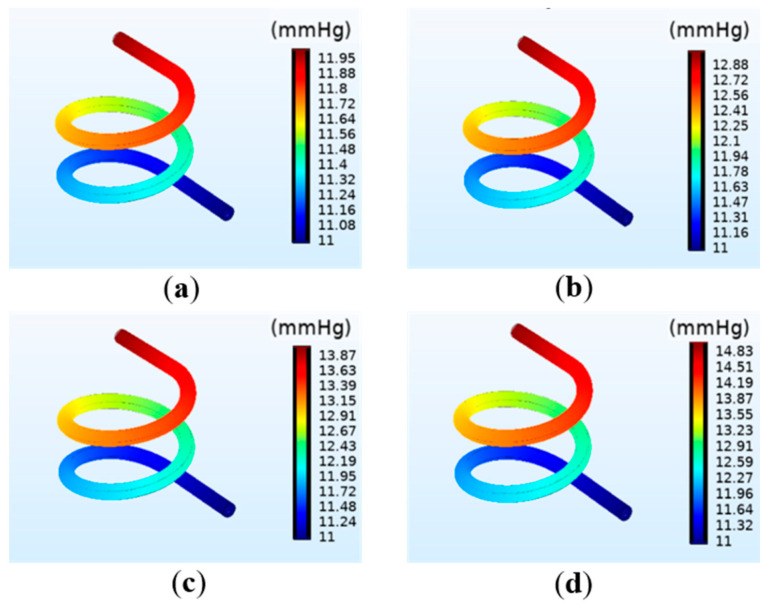
Simulation results of the vascular pressure under different perfusion pressure: (**a**) 12 mmHg; (**b**) 13 mmHg; (**c**) 14 mmHg; (**d**) 15 mmHg.

**Figure 5 polymers-12-01898-f005:**
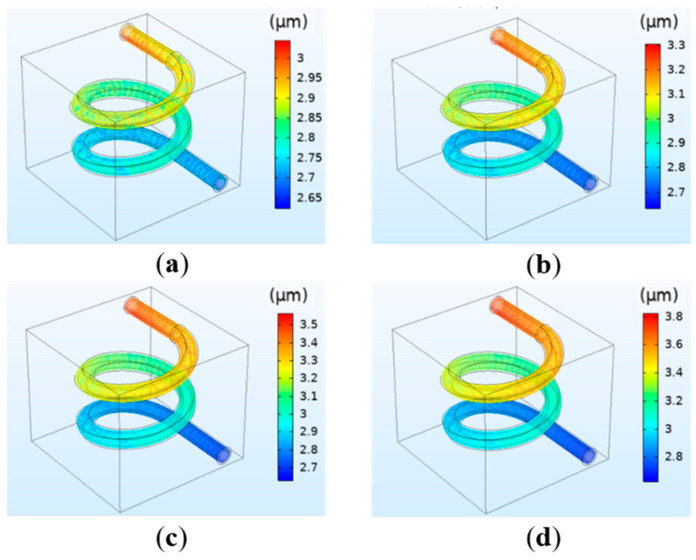
Simulation results of the vessel deformation under different perfusion pressure: (**a**) 12 mmHg; (**b**) 13 mmHg; (**c**) 14 mmHg; (**d**) 15 mmHg.

**Figure 6 polymers-12-01898-f006:**
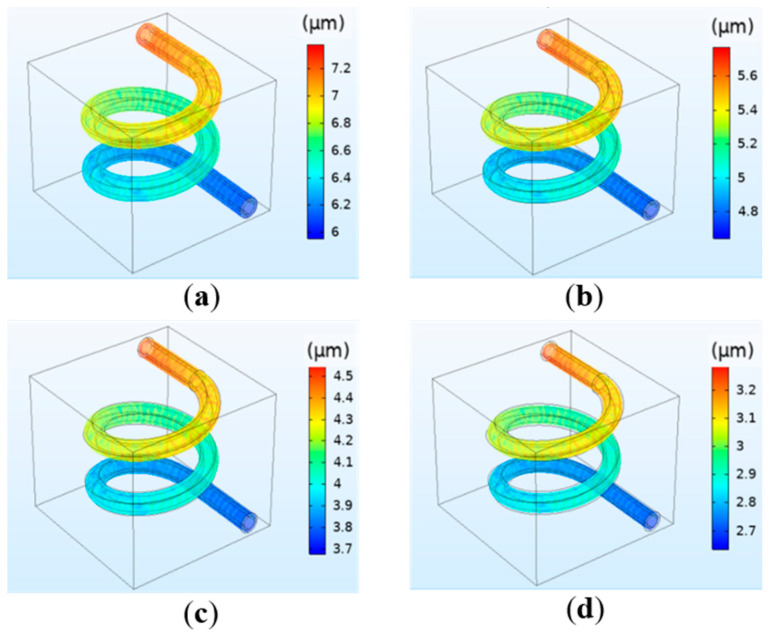
Simulation results of the vessel deformation when the perfusion pressure is fixed at 13 mmHg but the hydrogel concentration changes: (**a**) 20%; (**b**) 30%; (**c**) 40%; (**d**) 50%.

**Figure 7 polymers-12-01898-f007:**
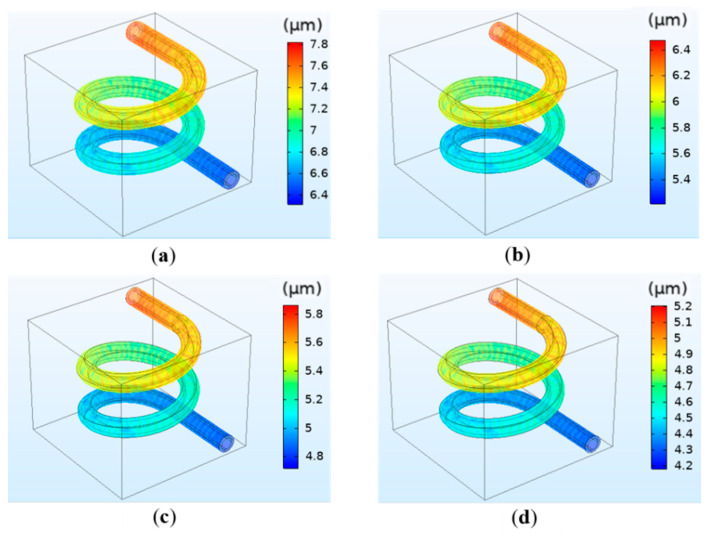
Simulation results of the vessel deformation when the crosslinking density is different: (**a**) 0.025 mol/L; (**b**) 0.05 mol/L; (**c**) 0.075 mol/L; (**d**) 0.1 mol/L.

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
