# Peer review of "Fluid–Structure Interaction Analysis of Perfusion Process of Vascularized Channels within Hydrogel Matrix Based on Three-Dimensional Printing"

_polymers, 2020, doi:10.3390/polym12091898_

Round 1
Reviewer 1 Report
Dear authors,
The manuscript provides simulation results of a method to optimize the
the perfusion parameters of tissue engineering, which is important for a successful in vitro cell culture towards tissue constructs. The manuscript seems quite sound, however, it is required a minor revisions prior to accepting for publication.
- The simulation includes the effect of sodium alginate hydrogel concentration on perfusion process. Generally, with increasing concentration of polymers, the elastic modulus of the hydrogel is increased. However, in case of hydrogels used for 3D cell culture in tissue engineering, crosslinking density is widely used to modulate the elasticity or to improve the mechanical properties of the hydrogels, rather than the concentration of polymers. As such, a curiosity pops up how the variation in crosslinking density of hydrogels impact the perfusion process. It is recommended to include crosslinking density in the simulation study, as well.
Thanking you,
sincerely,
Author Response
Dear reviewer,
Please see the attachment.
Thank you sincerely and best regards,
Shuai Yang

Reviewer 2 Report
In the manuscript the authors provide a method to optimize the flow perfusion parameters through a 3D printed biomaterial scaffold by performing fluid-structure interaction modelling. There are several points to be addressed.
- The text requires substantial grammar and English editing. For instance:
-lines 12-13: ‘to fabricate tissue engineering vitro’
-line 33: please correct the syntax of the following ‘The Since Wilson and Boland’
-line 37: ‘The culture of tissue engineering in vitro’. Do the authors mean ‘the in vitro culture of tissues?
-line 55: ‘The problem of how to further culture the printed tissue engineering in vitro is how to provide’. Please rephrase.
Line 61: ‘the metabolites produced in time’. Please correct the phrase in time
Line 63: ‘to solve the culture in vitro’. The sentence does not make sense
Lines 63, 69. 186: do authors mean scaffold instead of tissue engineering? If so, please change accordingly throughout the text.
Line 67: delete the word ‘the’
Line 79: add the word diagram after the schematic and correct the grammar of the sentence ‘..the process of fabricate..’
Line 81: correct the word immersing with immersion and the word removing with removal of
Line 92: correct the syntax ‘which made the model had better convergence’
Line 120: correct flow with flows
Line 121: ‘has a certain supporting effect’ define the word certain
Lines 145-146: please correct the syntax of the following ‘Due to the reference value of blood flow velocity of cerebral vein is 9.9±1.4 cm/s,’
Figure 3: the sentence ‘The results of fluid velocity when the perfusion pressure is’ should be removed and only the pressure value should be reported after (a), (b), (c) and (d).
The same for figures 4 and 5.
- The authors mention their previous work on developing perfusable vascularized channels but there is no citation of their work in the text. Please include the reference and provide more specific information of your previous work in the text.
- Why do authors perform hydrodynamic analysis of blood flow velocity and pressure since the perfusion medium in an in vitro vascularized tissue scaffold would be culture medium instead?
- The authors studied the deformation of the vessel from a materials perspective and under different perfusion pressures. However, the complex interaction between cells/tissues (e.g. in this case HUVEC cells) and the perfusion fluid flow is not taken into account. Please discuss on this.
Author Response

(The authors gave the same response as above.)

Reviewer 3 Report
The manuscript entitled "Fluid-structure interaction analysis of perfusion process of vascularized channels within hydrogel matrix based on three-dimensional printing" aims to demonstrate te possibility to use a simulation model able to investigate the effects of perfusion pressure and hydrogel concentration on the flow velocity. The topic is very interesting and could be helpful in bone tissue engineering field. However, minor revisions is required to improve the quality:
- Editing of English language is required;
- The Introduction section should be improved and the application field should be highlighted;
- It is interesting to investigate the effect of hydrogel degradation on the flow rate;
- the conclusion should be improved and in the first part the sentence "in the previous work" must be clarifed.
Author Response

(The authors gave the same response as above.)

Round 2
Reviewer 2 Report
All queries raised by the reviewer have been answered by the authors.